# Effectiveness of Dupilumab in the Treatment of Adult and Older Adult Patients with Severe, Uncontrolled CRSwNP

**DOI:** 10.3390/jpm13081241

**Published:** 2023-08-10

**Authors:** Giancarlo Ottaviano, Eugenio De Corso, Tommaso Saccardo, Leandro Maria D’Auria, Sonny Zampollo, Giuseppe D’Agostino, Edoardo Mairani, Gabriele De Maio, Bruno Scarpa, Christian Bacci, Vittorio Favero

**Affiliations:** 1Department of Neurosciences DNS, Otolaryngology Section, University of Padova, 35122 Padova, Italy; tommaso.saccardo@gmail.com (T.S.); sonny.zampollo@aopd.veneto.it (S.Z.); edoardomairani@gmail.com (E.M.); 2Department of Otolaryngology, Policlinico Gemelli University Hospital IRCCS, 00168 Rome, Italy; eugenio.decorso@policlinicogemelli.it (E.D.C.); leadauria1@gmail.com (L.M.D.); dagostinogiuseppe5@gmail.com (G.D.); gabrieledemaio1@gmail.com (G.D.M.); 3Department of Statistical Sciences, University of Padova, 35122 Padova, Italy; bruno.scarpa@unipd.it; 4Unit of Dentistry, Department of Neurosciences, University of Padua, 35122 Padova, Italy; christian.bacci@unipd.it; 5Unit of Maxillofacial Surgery and Dentistry, Head and Neck Department, University of Verona, 37134 Verona, Italy; vittorio_favero@yahoo.it

**Keywords:** chronic rhinosinusitis, elderly, younger–middle adults, Dupilumab, NPS, PNIF, Sniffin’ Sticks, VAS, Snot-22

## Abstract

Chronic rhinosinusitis with nasal polyps (CRSwNP) is a multifactorial disease that significantly impacts patients’ quality of life. New therapeutic strategies and in particular biologic treatments are now available for these patients. It has been demonstrated that Dupilumab (an anti IL-4/IL-13 biologic drug) is effective in reducing the size of nasal polyps and in improving patients’ symptoms and thus, quality of life. No real-world studies examining Dupilamab’s efficacy in the elderly with respect to other adult age groups have as yet been carried out. The aim of this multicentric study was to evaluate Dupilumab’s efficacy in young–middle adults as opposed to an older adult population affected by severe, uncontrolled CRSwNP. Of the 96 patients included in the study, 22 were 65 years old or older. Significant improvements were observed in all the parameters considered in both age groups after treatment was begun (T0 mean values for SNOT-22 = 58.5 ± 20.3, VAS NO = 7.6 ± 2.2, VAS smell = 8.6 ± 2.1, NPS = 5.6 ± 1.4, PNIF = 101.6 ± 59.4, S’S = 5.1 ± 3.1), T4 mean values for SNOT-22 = 15.1 ± 12.7, VAS NO = 1.7 ± 1.8, VAS smell = 2.4 ± 3, NPS = 1.7 ± 1.7, PNIF = 162.4 ± 43.2, S’S = 10.4 ± 3.7) (*p* < 0.0001). No differences in the variables considered were observed between the two age groups during the study, with the exception of the Peak Nasal Inspiratory Flow (PNIF), which was marginally higher; this was also the case according to multivariate analyses (*p* = 0.008) in the young–middle adult group with respect to the elderly one (*p* = 0.07). At multivariate analyses, asthma and the female sex negatively influenced the PNIF values (*p* = 0.001 and *p* = 0.012, respectively). Age negatively influenced the Visual Analog Scale (VAS) for nasal obstruction (*p* = 0.0032) and Endoscopic Sinus Surgery (ESS) negatively influenced the patents’ olfactory performance (*p* = 0.028) to the same degree in both groups. Dupilumab was found to be effective to the same degree in both age groups. It can be considered a safe and reliable option for the treatment of elderly patients with severe, uncontrolled CRSwNP.

## 1. Introduction

According to the latest projections, the proportion of the population that is over 65 in Western countries is growing, and it is estimated that over 20% of the population will be over 65 by 2050 in these areas [1]. As the proportion of the population over 65 is increasing, so are the challenges in diagnosing and managing some pathologies in elderly adults. Chronic rhinosinusitis (CRS) is a common condition in elderly people. It may be associated with important functional problems, such as nasal obstruction, particularly when nasal polyps are present. Chronic rhinosinusitis with nasal polyps (CRSwNP) is known to ultimately reduce the quality of life of affected patients. The incidence of CRSwNP, which is higher in men, is reported to be approximately 1–4% of the general population, with ~5% of elderly adults suffering from CRSwNP [1] CRSwNP’s disease course is highly variable as some patients present with rare mild symptoms, whereas in others these are frequent, causing discomfort and affecting quality of life.

Causing nasal obstruction, nasal discharge, loss of taste and smell, toothache, ear pain, and or headaches, generally speaking, CRSwNP reduces patients’ overall health and quality of life [1,2]. According to international guidelines, diffuse, bilateral CRS should be treated with local corticosteroids and saline irrigations, but these need to be delivered to compliant patients using appropriate techniques. If intranasal corticosteroids and saline treatments are insufficient to control the disease, oral steroids should be added. However, the side-effects of these drugs, including negative cognitive and psychiatric effects and memory issues [3], and the fact that patients at high risk of complications should not be treated with systemic steroids are important considerations [4]. In the absence of significant comorbidities, i.e., diabetes, blood hypertension, osteoporosis and glaucoma, two short courses of systemic corticosteroids should be prescribed per year. If oral therapy fails, endoscopic sinus surgery (ESS) should be considered, especially in those cases in which no previous sino-nasal surgery has been offered [4].A biologics-based therapy has recently been utilized to manage CRSwNP in the light of current knowledge regarding the disease’s underlying pathophysiological mechanisms and a precision medicine approach. In fact, precision medicine, which tailors treatment to the patients’ individual characteristics and response to a therapy, has a higher probability of achieving improved outcomes [5]. Several works have reported that Dupilumab (an anti-IL-4/IL-13 biologic drug) effectively reduced the size of nasal polyps and improved the important parameters of disease burden in a number of patients [6,7].

Chronic rhinosinusitis, which is the sixth most common chronic condition in the elderly, can recur following sino-nasal surgery, particularly in patients with comorbidities such as allergies, asthma, and non-steroidal anti-inflammatory drugs (NSAID) intolerance [8]. The surgical risk in fragile patients, such as elderly subjects, is an important consideration. A more personalized treatment based on biologics rather than additional revision surgery may be preferable, particularly in patients with comorbidities. No studies have as yet been designed to compare Dupilumab’s efficacy in young–middle adult patients with severe, uncontrolled forms of CRSwNP as opposed to that in an older adult population.

The aim of this retrospective multicentric study was to evaluate the effects of Dupilumab in a group of elderly patients with severe, uncontrolled CRSwNP and to compare them with those in a comparable group of young–middle adult patients. The prognostic factors for CRSwNP and the demographic, clinical, laboratory, and prognostic characteristics of this condition in these populations were also analyzed.

## 2. Materials and Methods

A retrospective multicentric study involving two important Otolaryngology centers located in Italy were designed to evaluate patients affected by severe uncontrolled CRSwNP. The patients studied fell into two groups: young–middle adult patients, all under 65, and older patients, all over 65. The patients were attending the Rhinological Unit of the Padua University Hospital and the Rhinology Unit of the Gemelli Hospital Foundation-IRCCS, the Catholic University of the Sacred Heart of Rome.

All of the patients attending the two Rhinology Units who were diagnosed with severe and uncontrolled CRSwNP in accordance with the criteria of the European Position Paper on Rhinosinusitis and Nasal Polyps (EPOS) 2020 [4] and taking 300 mg of Dupilumab every 15 days in addition to intranasal corticosteroids were evaluated for eligibility to the study. All patients included in the study were regularly using nasal steroids before and throughout the whole study period. Used molecules were either Mometasone furoate 50 μg 2 sprays in each nostril once or 2 times a day or budesonide 100 μg 1 spray in each nostril once a day. Asthmatic patients were regularly using a combination of long-acting beta agonist (LABA) and inhaled corticosteroids (ICS) and never discontinued the treatment.

The Italian Medicine Agency recommends prescribing Dupilumab to CRSwNP patients over 18 showing inadequate symptom control despite the use of intranasal corticosterioids who had already received two or more cycles of systemic corticosteroids over the past year or who had undergone ESS. The agency uses the following criteria to define severe CRSwNP: a nasal polyp score (NPS) > 5 and/or a Sinonasal Outcome Tests-22 (SNOT-22) > 50.

The study was carried out in compliance with all of the principles of the Declaration of Helsinki and in conformity with the approval of each unit’s own ethics committee (Padua University Hospital: 53054/AO/22. Gemelli Hospital of Rome: ID 4429).

All of the patients who participated in the study were given a detailed description of the study’s aims and modality and were asked to sign consent statements.

Data at the time of baseline (before the biological treatment was begun) (T0) and during all the follow-up visits, which were held one month after starting treatment (T1), 3 months after starting treatment (T2), 6 months after starting treatment (T3), and 12 months after starting treatment (T4), were collected. The patients underwent nasal endoscopy using a 0° and or 30° rigid endoscope and the Nasal Polyp Score (NPS) was scored in accordance with the indications of Gevaert et al. at the baseline and at every follow-up examination [9]. The patients’ quality of life was evaluated using the Sino-Nasal Outcome Test (SNOT-22) questionnaire [10]. The severity of nasal symptoms was measured subjectively using the Visual Analogue Scale (VAS) for nasal obstruction (NO) and for olfactory impairment [11] and objectively using the Peak Nasal Inspiratory Flow (PNIF) meter (Clement Clark International, Mountain Ash, UK) [12] and the Sniffin’ Sticks Identification test (SSIT) (16 odors) (Burghart Messtechnik GmbH, Holm, Germany) [13].

The criteria for patient exclusion were: eosinophilic granulomatosis with polyangiatis (EGPA), cystic fibrosis, pregnancy, radiochemotherapy for cancer during the 12 months before the treatment was to begin, or concomitant long-term oral corticosteroid therapy for chronic autoimmune disorders.

### Statistical Analysis

Sample descriptive statistics were obtained to analyze the effect over time of the variables of interest; in particular, mean, standard deviation, and quantiles were computed for quantitative variables and absolute and relative frequency distributions for categorical ones.

Inference was implemented through the longitudinal analysis of all the relevant clinical quantities, such as PNIF, SSIT, SNOT-22, VAS-NO (nasal obstruction), VAS-smell, and NPS, and for each of these quantities, linear mixed models were fitted to the data using as covariates asthma, non-steroidal anti-inflammatory drugs (NSAID) intolerance, allergy, smoking habit, previous ESS, and number of previous surgeries. In one of our group’s previous works [14], we showed that the heterogeneity in the variability of PNIF can be reduced through a square root transformation MODPNIF = (PNIF)1/2; therefore, in accordance with this, we modeled MONDPNIF instead of PNIF. No transformation was judged necessary for the other clinical variables. We observed that the effects of all the clinical quantities analyzed followed a nonlinear trend over time, therefore, we decided to include a different parameter for each time in all models. The significance of the effects over time and of all the covariates was obtained using the analysis of variance tables and Snedecor F-tests. The denominator degrees-of-freedom of the F-statistic were calculated using Satterthwaite’s method [15], and consequently, the observed significance levels (*p*-values) were computed. The selection of the best multiple predictive model was performed through a backward stepwise procedure based on the Akaike Information Criterion (AIC) [16]. The different complexities of the models, in addition to the goodness of fit, were also considered. The same AIC criterion was also used to compare the different best models obtained. The best-fit model, according to AIC, is the one characterized by the lowest AIC value. Observed significance levels (*p*-values) were obtained for all the tests, and 5% was considered the critical level of significance; *p*-values between 0.05 and 0.10 were considered a trend towards significance.

Multiple regression was fitted to consider all the variables examined (sex, age, smoking, asthma, endoscopic sinus surgery, the number of oral corticosteroids cycles over the past year, allergies, and NSAID intolerance or Widal’s disease triad) in a mixed-effects model to test the significance of the differences in treatment responsiveness and clinical outcomes between older and young–middle-aged subjects over time.

R, a language and environment for statistical computing (R Foundation for Statistical Computing, Vienna, Austria), was used for all the analyses [17].

## 3. Results

A total of 96 patients (64 males and 32 females, mean age 51.2 ± 13.4 years) who received Dupilumab therapy for at least one year for severe, uncontrolled CRSwNP were considered for this multicentric study. Seventy-four of the patients fell under the young–middle adult classification, meaning that they were younger than 65. Twenty-two of the patients fell under the older adult classification, meaning that they were older than 65. The patients’ most relevant characteristics at baseline (T0) are listed in Table 1.

Beginning at the first follow-up visit after Dupilumab was prescribed (T1), a significant improvement in sino-nasal symptoms was noted in both patient groups (based on the SNOT-22, VAS for nasal obstruction and the VAS for smell values); there was also a significant decrease in the nasal polyps’ size, an improvement that was confirmed throughout the study period (*p* < 0.0001) (Figure 1, Figure 2, Figure 3 and Figure 4).

There were no differences in these variables in the two age groups. The nasal airflow measured using the PNIF method was higher in both groups at T1 with respect to the baseline (T0) value, and the improvement was confirmed throughout the 12-month follow-up period (T2, T3 and T4) (*p* < 0.0001); there was a marginally greater improvement in the young–middle adult group with respect to the elderly one (*p* = 0.07) (Figure 5). The sense of smell measured using the SSIT also improved significantly after Dupilumab therapy was begun, and the improvement was confirmed throughout the 12-month follow-up period (T1, T2, T3, T4) (*p* < 0.0001) in both groups (Figure 6).

The multivariate analysis conducted using a model that included all the variables examined to assess their effect on the patients’ response to Dupilumab therapy confirmed that the PNIF was significantly lower in the elderly group with respect to that in the other group (*p* = 0.008). In addition, at the multivariate analysis asthma, and female sex were found to have a negative effect on the PNIF values (*p* = 0.001 and *p* = 0.012, respectively). With regard to the VAS for nasal obstruction, age was found to have a significant negative effect on that score at the multivariate analyses (*p* = 0.0032) (Table 2), although there were no significant differences between the two groups studied. Finally, with regard to the SSIT, previous EES had a negative effect, according to the multivariate analyses (*p* = 0.028), but no differences between the two age groups were noted (Table 3).

Differences between males and females and smokers and non-smokers are summarized in Table 4 and Table 5.

No adverse events were observed during the treatment period. We performed blood samples once a month to monitor the blood eosinophils levels. There was a transient increase in blood eosinophils in many of the patients, but values consistent with hypereosinophilia (eosinophils >1.5 × 10^9^/L) were found in only eight out of the ninety-six patients studied. With the exception of one patient who developed arthralgia and needed to suspend the therapy, all of the patients with hypereosinophilia experienced a rapid spontaneous resolution within 3 months, not requiring oral corticosteroid treatment nor needing to discontinue the biological therapy. Of the nine patients developing transient hypereosinophilia, only two fell under the elderly group classification. None of the patients in either group developed conjunctivitis [18]. None of our patients took systemic corticosteroids for either asthma or CRSwNP exacerbations after the beginning of Dupilumab throughout the first year of follow-up.

## 4. Discussion

Dupilumab has been approved for five indications: moderate-to-severe atopic dermatitis, moderate-to-severe eosinophilic or oral corticosteroid-dependent asthma, eosinophilic esophagitis, prurigo nodularis, and inadequately controlled chronic rhinosinusitis with nasal polyps. The findings of the current work confirm those presented by other real-life studies [19] demonstrating that Dupilumab is associated to a significant improvement in sino-nasal symptoms (measured using SNOT-22, VAS for nasal obstruction and VAS for smell) and objective measurements (PNIF and SSIT). As outlined above, the results collected over the study period from both age groups were similar, and we can thus conclude that a significant improvement in symptoms and in disease control can be obtained in elderly patients independently of the variables associated with a type-2 inflammation such as atopy, NSAID intolerance, previous endoscopic sinus surgery, the number of cycles of systemic corticosteroids undertaken over the last year, and the presence of asthma. Looking at the intragroup and intergroup changes during biological therapy, it seems clear that the effects of the therapy were similar in both groups, with changes much more evident in each group during the follow-up than between the groups. Table 2 and Table 3 clearly show that for both VAS-NO and SSIT, the effects of the therapy during the time (time 1–3–6–12 months) (intragroup effect) were much higher than the effects of age (intergroup effect). For VAS-NO in particular, at the baseline, the mean difference between the elderly patients and the young–middle adult group was −0.036 (intergroup effect), whereas the mean VAS-NO difference just after the first month of follow-up (T1) was −4.26 (intragroup effects) (Table 2). A very similar result is shown in Table 3 for SSIT. In particular, we could not find an intergroup effect due to age between young–middle adults and the elderly for SSIT, whereas there was a strong intragroup effect due to the therapy (3.73 in the first month of treatment and 5.3 after the 12th month of treatment) (Table 3).

An analysis of study findings uncovered that there was a marginally significant lower PNIF value in the elderly group with respect to their younger counterparts. We can explain this result, which was confirmed by the multivariate analyses, by the fact that age is one of the main factors affecting PNIF values [14,20]. In particular, elderly people have a physiological reduction of nasal airflows (i.e., PNIF [20]) and increased nasal resistances [21]. The reduced nasal airflow in the elderly could be a consequence of the physiological limitation of the pulmonary function that has been shown in this age group [22], as nasal airflows and pulmonary volumes are strictly linked [23]. Furthermore, in the elderly, there is physiological cartilage weakening that can lead to a reduction in the nasal valve area due to a nasal alar collapse and nose tip drop [24]. The strict correlation that exists between the upper and the lower airway volumes can also justify the negative influence that asthma had on PNIF values in the multivariate analyses. Finally, the multivariate analyses revealed that female sex had a negative effect on the PNIF values. This could be due to the fact that, in previous studies conducted in a normal population, females were shown to have lower PNIF values with respect to their male counterparts [14,24].

The findings of cross-sectional studies suggest that about half of the US population between the ages of 65 and 80 has a demonstrable olfactory impairment. Among individuals over 80, approximately three-quarters experience olfactory loss [9]. Moreover, it has been demonstrated that aging is significantly associated with a reduced ability to identify odors by name (identification test), and the elderly may also experience a significant loss of olfactory perception [25]. It has also been shown that the elderly have higher olfactory thresholds with respect to young adults [26]. The etiology of age-related olfactory loss is unclear. Aging itself may cause changes in the olfactory epithelium, including reduced mucus secretion and even olfactory neuron loss due to the neurons’ diminished ability to regenerate. Moreover, the number of glomeruli in the olfactory bulb also decreases [25].

No differences were noted in the two groups as far as the odor identification ability, measured by means of the SSIT, was concerned. Interestingly, the multivariate analyses uncovered that the SSIT values were negatively influenced by previous sinus/nasal surgery in both age groups. This result seems to confirm that sinus surgery, especially when it involves the olfactory cleft, can cause irreversible damage to the sense of smell. Brinner et al., in fact, reported uncovering a smell impairment in about 15% of patients treated with endoscopic sinus surgery for chronic rhinosinusitis [27], a finding that has been confirmed by other investigators [28]. We would have expected to find olfactory function deterioration in the elderly group due to the physiological changes associated with aging [29]. It is possible that those changes are masked in CRSwNP patients by an olfactory dysfunction due to the persistent inflammation of the mucosa of the nasal cavity and the sinuses. In the future, the availability of a larger number of alternative molecules in addition to a reduction in the treatment costs might lead to an extension to the current indications to biologic treatment and improve our knowledge on which biological therapy will be more suitable for elderly patients. At the moment, from the findings of the present study, Dupilumab seems to be a good choice for the treatment of elderly patients with severe and uncontrolled CRSwNP.

The present investigation has its limitations. Although our study was conducted in two rhinological centers that are well recognized in Italy with known experience in the treatment of severe and uncontrolled CRSwNP patients, we were only able to recruit a relatively small number of elderly patients, which, for example, prevented any possible comparison between different subtypes of CRSwNP. Increasing the number of individuals in the elderly patients’ group could also allow for the evaluation of whether sub-groups of elderly patients based on different subtypes of CRSwNP could have different responses to Dupilumab. The present work also lacks a cytological study of the nasal smears. The absence of the local tissue inflammation evaluation could be considered another limitation of the study. Recently, a cytological evaluation of the nasal smears in severe and uncontrolled CRSwNP patients treated with Dupilumab was able to demonstrate a reduction of the eosinophils as well as of the neutrophils in the nasal smears [7]. It seems that neutrophils are major inflammatory cells in refractory CRSwNP, with the mucosal inflammation being dominated by neutrophils at least in a third of the patients with severe forms, in a mixed population [7]. The action of Dupilumab against the neutrophilic local inflammation can be a key action to treat and control CRSwNP patients, also in the elderly. In future, this local inflammatory response to Dupilumab could be evaluated in elderly patients to verify if similar findings can be obtained. Finally, unilateral PNIF measurements could increase our knowledge about the effect of Dupilumab in the nasal airflows [30].

## 5. Conclusions

It goes without saying that since CRSwNP is not a life-threatening condition [31], elderly patients’ comorbidities [32] and anesthetic risk need to be considered carefully, particularly in light of the evidence demonstrating that elderly patients undergoing EES have significantly higher rates of complications [33,34,35]. The findings reported here demonstrating that Dupilumab has basically the same effect in both the younger–middle adults and older adults indicate that it can be considered a safe, reliable option with respect to the surgical treatment of elderly patients, especially those who are at high surgical risk and or those at high risk of surgical failure due to nasal polyps relapse (i.e., elderly patients with asthma, atopy, and those with nonsteroidal anti-inflammatory drug (NSAID)-exacerbated respiratory disease). This is particularly true for the latter in particular, being that in this case, the CRSwNP is typically more extensive and more recalcitrant to medical and surgical treatments [36].

In future, larger studies, preferably within larger multicenter settings, are warranted in order to allow for the stratification of the population by phenotypes/endotypes and the quantification, also by means of a study of the local inflammation (i.e., using nasal cytology [37]), of any difference in treatment response between the adult and the older adult patients and between the different sub-groups of patients.

## Figures and Tables

**Figure 1 jpm-13-01241-f001:**
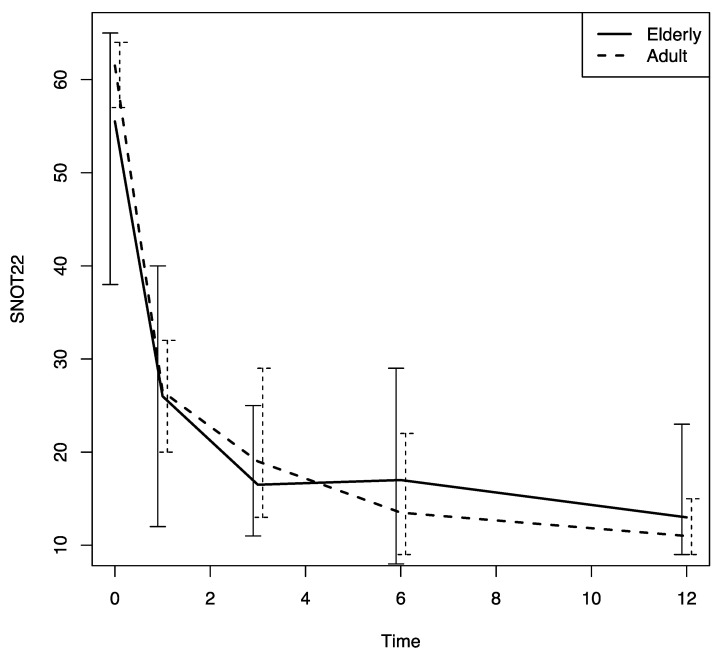
SNOT-22 changes in both groups during the study follow-up period. SNOT-22: sinonasal outcome test-22; time: months.

**Figure 2 jpm-13-01241-f002:**
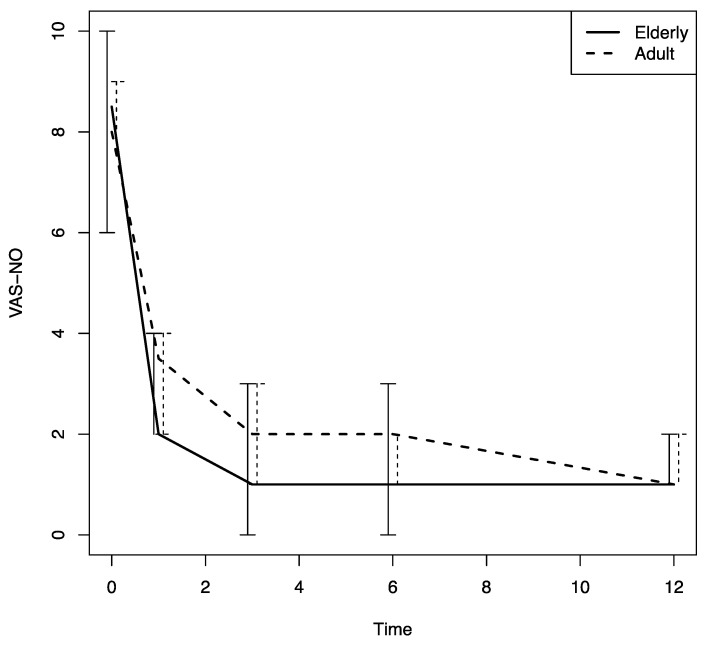
VAS-NO changes in both groups during the follow-up. VAS-NO: visual analogue scale for nasal obstruction; time: months.

**Figure 3 jpm-13-01241-f003:**
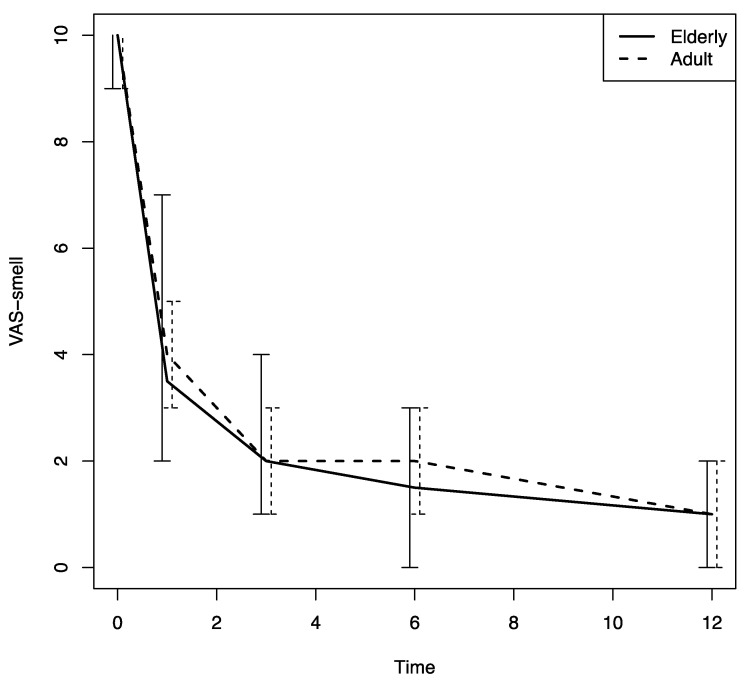
VAS-smell changes in both groups during the follow-up. VAS-smell: visual analogue scale for smell; time: months.

**Figure 4 jpm-13-01241-f004:**
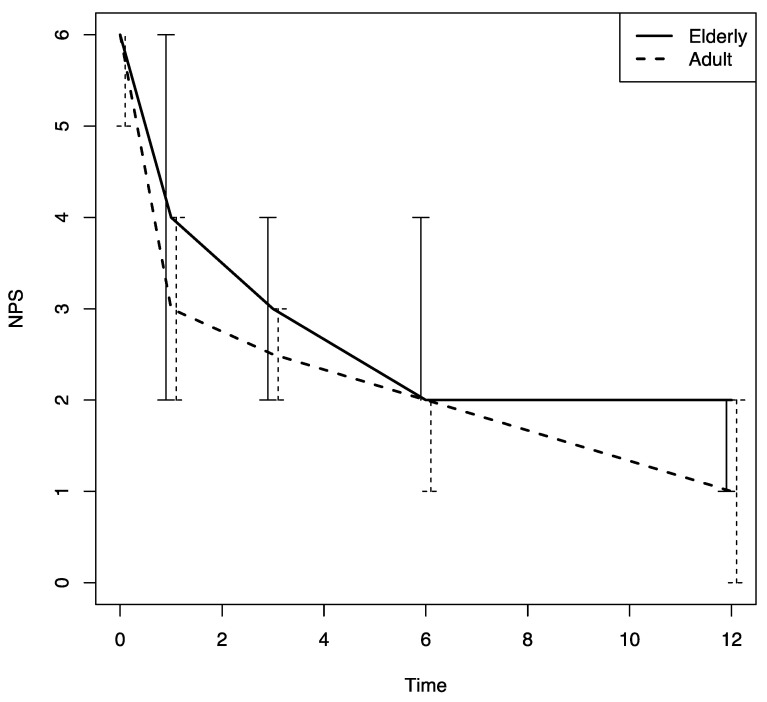
NPS changes in both groups during the follow-up. NPS: nasal polyp score; time: months.

**Figure 5 jpm-13-01241-f005:**
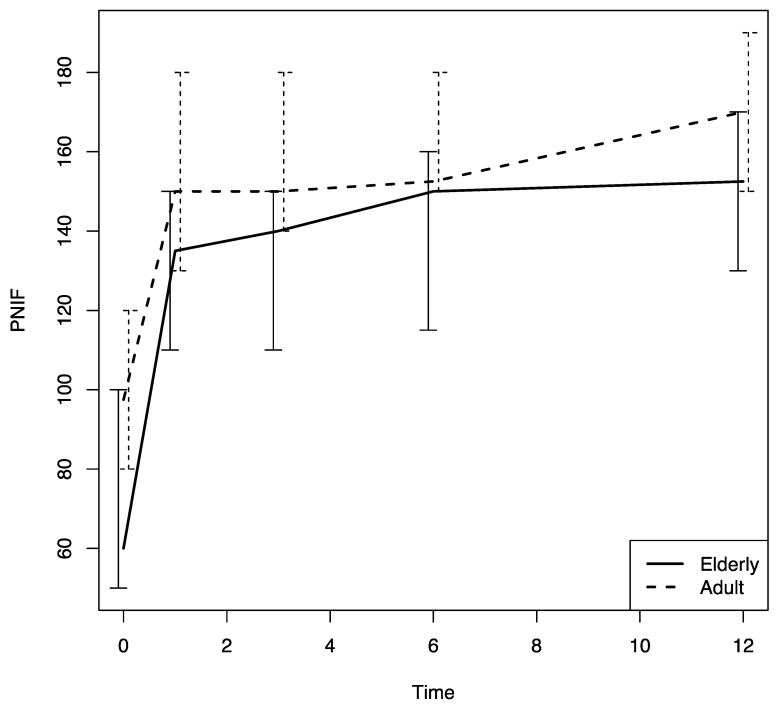
PNIF changes in both groups during the follow-up. PNIF: peak nasal inspiratory flow; time: months.

**Figure 6 jpm-13-01241-f006:**
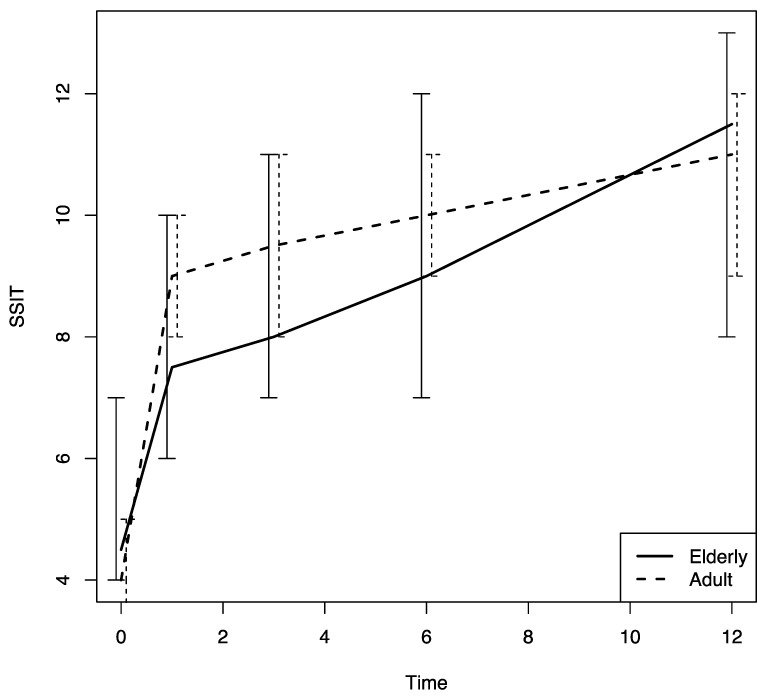
SSIT changes in both groups during the study follow-up period. SSIT: sniffin’ sticks; time: months.

**Table 1 jpm-13-01241-t001:** The most relevant clinical characteristics at baseline of the patients participating in the study.

	ALL *n* = 96	Adult *n* = 74	Elderly *n* = 22
Sex	32 Women 64 Men	25 Women 49 Men	7 Women 15 Men
Mean Age, yr (SD)	51.2 (13.4)	46.1 (10.7)	68.8 (4.1)
Asthma, *n* (%)	60 (62.5)	45 (60.8)	15 (61.8)
NSAID intolerance, *n* (%)	23 (24)	18 (24.3)	5 (22.7)
Allergy, *n* (%)	60 (62.5)	46 (62.2)	14 (63.6)
Smokers, *n* (%)	25 (26)	20 (27)	5 (22.7)
Previous ESS, *n* (%)	91 (94.8)	70 (94.6)	21 (95.6)
Mean *n*. of previous surgeries, *n* (SD)	1.83 (0.91)	1.75 (0.90)	1.68 (0.95)
Mean *n*. of short OCS cycles, *n* (SD)	2.87 (2.67)	3.06 (2.96)	2.22 (1.15)

*n*: number; yr: years; SD: standard deviation; NSAID: non-steroidal anti-inflammatory drugs; ESS: endoscopic sinus surgery; and OCS: oral corticosteroids.

**Table 2 jpm-13-01241-t002:** Multivariate linear regression analysis (linear mixed model) to assess the variable’s effect on VAS-NO. The last columns show the estimates of the effects and relative *p*-values for the univariate models.

	Multivariate Model		Univariate Models	
	Estimate	Estimate	Estimate	*p*-Value
**Intercept**	9.52336	<0.0001		
**Age**	−0.03654	0.0032	−0.036	** 0.0007 **
**T1**	−4.26042	<0.0001	−4.260	** <0.0001 **
**T3**	−5.51042	<0.0001	−5.510	** <0.0001 **
**T6**	−5.60417	<0.0001	−5.604	** <0.0001 **
**T12**	−6.00000	<0.0001	−6.000	** <0.0001 **
***Random effect*:**	** *variance* **	** * std. error * **	−6.000	** <0.0001 **
**patient**	1.873	1.369		
**Residuals**	2.970	1.723		

Statistically significant values (*p* < 0.05) are marked in bold. VAS-NO: Visual analogue scale for nasal obstruction. T1: one month after starting treatment; T3: three months after starting treatment; T6: six months after starting treatment; and T12: twelve months after starting treatment.

**Table 3 jpm-13-01241-t003:** Multivariate linear regression analysis (linear mixed model) to assess the variable’s effect on SSIT. The last columns show the estimates of the effects and relative *p*-values for the univariate models.

	Multivariate Models		Univariate Models	
	Estimate	*p*-Value	Estimate	*p*-Value
**Intercept**	7.4746	** <0.0001 **		
**ESS**	−2.5007	** 0.0284 **	−2.501	** 0.0017 **
**Time 1 month**	3.7292	** <0.0001 **	3.729	** <0.0001 **
**Time 3 months**	4.0938	** <0.0001 **	4.094	** <0.0001 **
**Time 6 months**	4.3021	** <0.0001 **	4.302	** <0.0001 **
**Time 12 months**	5.3021	** <0.0001 **	5.302	** <0.0001 **
***Random effect*:**	** *variance* **	** * std. error * **		
**patient**	4.552	2.134		
**Residuals**	7.127	2.670		

Statistically significant values (*p* < 0.05) are marked in bold. SSIT: Sniffin’ Sticks Identification test. T1: one month after starting treatment; T3: three months after starting treatment; T6: six months after starting treatment; and T12: twelve months after starting treatment.

**Table 4 jpm-13-01241-t004:** Shows the differences between males and females for the variables SNOT 22, VAS-NO, SSIT, and NPS.

	Median M	Median F	*p*-Value *
**SNOT22 T0**	57.0	64.0	**0.021**
**SNOT22 T1**	26.5	26.0	0.494
**SNOT22 T3**	17.0	19.0	0.323
**SNOT22 T6**	15.0	14.5	0.663
**SNOT22 T12**	12.0	10.5	0.981
**VAS-NO T0**	8	8	0.342
**VAS-NO T1**	3	2.5	0.513
**VAS-NO T3**	2	2	0.634
**VAS-NO T6**	2	1	0.308
**VAS-NO T12**	1	1	0.660
**SSIT T0**	5.0	4.0	0.151
**SSIT T1**	9.0	9.0	0.966
**SSIT T3**	9.0	9.5	0.953
**SSIT T6**	10.0	10.0	0.907
**SSIT T12**	10.0	12.0	0.571
**NPS T0**	6.0	6.0	0.356
**NPS T1**	3.0	3.0	0.512
**NPS T3**	3.0	3.0	0.741
**NPS T6**	2.0	2.0	0.351
**NPS T12**	1.0	1.5	0.971

SNOT22: sinonasal outcome test-22; VAS-NO: visual analogue scale for nasal obstruction; SSIT: Sniffin’ Sticks Identification test; and NPS: nasal polyp score. M: males; F: females. T1: one month after starting treatment; T3: three months after starting treatment; T6: six months after starting treatment; and T12: twelve months after starting treatment. * Mann–Whitney test.

**Table 5 jpm-13-01241-t005:** Shows the differences between smokers and non-smokers for the variables SNOT 22, VAS-NO, SSIT, and NPS during the study between males and females and between smokers and non-smokers.

	Median Smoking	Median No Smoking	*p*-Value *
**SNOT22 T0**	58	63	0.387
**SNOT22 T1**	27	25	0.442
**SNOT22 T3**	17	23	0.805
**SNOT22 T6**	15	17	0.713
**SNOT22 T12**	13	10	0.373
**VAS-NO T0**	8	8	1
**VAS-NO T1**	2	4	1
**VAS-NO T3**	2	2	1
**VAS-NO T6**	2	1	1
**VAS-NO T12**	1	1	1
**SSIT T0**	5	4	0.498
**SSIT T1**	9	9	0.451
**SSIT T3**	9	9	0.864
**SSIT T6**	9	10	0.201
**SSIT T12**	11	11	0.651
**NPS T0**	6	6	0.802
**NPS T1**	3	3	0.919
**NPS T3**	3	2	0.748
**NPS T6**	2	2	0.782
**NPS T12**	2	1	0.404

SNOT22: sinonasal outcome test-22; VAS-NO: visual analogue scale for nasal obstruction; SSIT: Sniffin’ Sticks Identification test; NPS: nasal polyp score. T1: one month after starting treatment; T3: three months after starting treatment; T6: six months after starting treatment; and T12: twelve months after starting treatment. * Mann–Whitney test.

## Data Availability

The data presented in this study are available on reasonable request from the corresponding author.

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
