# Peer review of "Effectiveness of Dupilumab in the Treatment of Adult and Older Adult Patients with Severe, Uncontrolled CRSwNP"

_jpm, 2023, doi:10.3390/jpm13081241_

Round 1

Reviewer 1 Report

The aim of this multicentric study was to evaluate the effects of Dupilumab in a group of elderly patients with severe uncontrolled CRSwNP and to compare them with those in a comparable group of young-middle adult patients. 

The sudy is well designed, well studied, providing a guiding contribution to practical medicine. My criticisms are as follows; 

1. The following paragraphs should be included in the introduction section instead of the conclusions. Since the situation of the elderly patients are already mentioned in the introduction, while carrying the knowledge, the author should pay attention in order to not double the knowledge. 

 "According to the latest projections, the proportion of the population that is over 65 in Western countries is growing, and it is estimated that over 20% of the population will be over 65 by 2050 in these areas [1]. Chronic rhinosinusitis is a common condition in elderly people, and the rate of nasal polyps is significantly higher in elderly patients [26]. The negative impacts of CRSwNP are multiple including nasal discharge, loss of taste and smell, toothache, ear pain and or headache and generally speaking it reduces patients’ overall health and quality of life [1].

Besides intranasal corticosteroids, short courses of oral corticosteroids are a valuable component of the medical management of CRSwNP, even in the elderly [27]. The side effects such as osteoporosis, diabetes and hypertension, should however be considered, especially in elderly, polymorbid patients [3]. While short steroid cycles can have a positive effect on the patient’s condition, they have also been linked to negative cognitive and psychiatric effects, including memory issues [28] and need to be used judiciously on an individualized basis. When medical therapy fails, EES can be indicated [3]. "

2. The discussion about the effect of treatment on PNIF in elderly patients is not clear. Authors should clearly stress how is "in the group changes" and "intergroup changes" before and after the treatment.  

Author Response

 According to reviewer 1 suggestions:

  1. The first two paragraphs of the conclusion section have been removed and the introduction modified to avoid duplications. Some references have been removed and the reference list has been updated.
  2. The discussion about the effect on PNIF in the elderly have been enlarged to improve clarity (page 15, lines 314-321)
  3. As suggested, we evaluated the intergroup (adults and elderly) and the intragroup effects of the therapy with dupilumab finding that the effects of the biological therapy were much higher than the effects of age. As an example, in table 2 and 3 for VAS-NO and SSIT it is possible to observe a much higher intragroup effect than an intergroup effect. We added these comments in the discussion (see page 14, lines 299-310)

Reviewer 2 Report

The authors have evaluated retrospectively, the effect of dupilumab on CRSwNP in patients from two age groups with follow-up information available for one year after initiation of the treatment. The study is fairly straightforward and adds important information.

General comments: spelling and grammar needs to be corrected at several sections of the manuscript and abstract

Abstract: Line 21-22: please modify the sentence, not clear

Manuscript:

Line 47-49: modify the sentence, not clear

Line 153: modify, the meaning is not clear.one suggestion:  (differences in treatment responsiveness and clinical outcomes between older and young-middle-aged subjects over time.)

Please present the full model for Table 2 and Table 3 including those variables that have not been identified as significant and please present both univariate and multivariate analysis.  

Figures: 

please present 95% CI for each of the time points in all the figures

please present a sub-group analysis to assess differences between gender for the SNOT 22,  VAS nasal obstruction, loss of smell, nasal polyp score

Similarly, are there differences between smokers and non-smokers in their response to dupilumab.

Conclusion:

please shorten the conclusion. Some of the material can be shifted to the section before the conclusion as future directions. Generally, conclusions are limited to information generated from the author's study.

There are several areas where language needs to be edited. Some examples are given above

Author Response

 According to reviewer 2 suggestions:

  1. The English has been revised through the text and the lines 21-22, 47-49, 153 have been modified to make the sentences more clear.
  2. the full statistical model, which includes the estimates of times, has been added in both tables (see tables 2 and 3). We also included in both tables the estimates of the effects and the p-values in the univariate models of age and time for VAS-NO and ESS and time for SSIT. In the tables all the variable available for the models and selected through a backward stepwise procedure based on the Akaike Information Criterion have been added.
  3. We included in every figure the 95% confidence intervals for the medians in the observed times, both for adult and elderly patients.
  4. To try to accomplish with the reviewer suggestion, two different tables have been done with sub-group analysis to assess differences for the variables SNOT 22, VAS nasal obstruction, olfactory test (SSIT) and nasal polyp score during the study between males and females (table 4) and between smokers and no smokers (table 5).
  5. The conclusion has been shortened, as requested.

Reviewer 3 Report

This is a study of dupilumab on nasal polyps in 96 patients. Dupilumab was found as safe treatment for severe uncontrolled CRSwNP in both the young-middle aged (younger than 65 years of age) and the elderly group of patients (65 years or older). PNIF values were higher in the younger ones and lower in females and in those with asthma. VAS was lower in the elderly and olfactory results were lower in those who had had endoscopic sinus surgery. The study is important in it’s perspective to analyze the effect of dupilumab in the elderly patients which is often underestimated in real life.

Major

-        What are the main results of the study? Please, express them in numerical values in the abstract.

-        In the Methods section it is described that all the adult patients with severe CRSwNP taking nasal corticosteroids and dupilumab every 15 days were evaluated for the study. On the other hand, baseline information of the symptoms and the polyp score results were gathered. This seems to be a retrospective study if all the patients with dupilumab were evaluated for the study. Please, clarify this in the Method section.

-        Is there information on the use of nasal steroids (which of the nasal corticosteroids, what was the dose and what was the compliance to use it?)

-        Is there information on the use of per oral glucocorticoids before and during the study?

-        Is there information on the use of asthma drugs (inhaled glucocorticoids, montelukast, per oral glucocorticoids) before and during the study?

-        Please explain the meaning of VAS-NO in the Methods section.

-        Table 2 and Table 3. Please, consider giving the used variables in the Title of the tables. The Reader needs to return to the Methods section to understand what variables were used in the linear regression models (linear mixed models) when estimating the Tables.

-        How quickly did the eosinophilia resolve in the patients? (In the Results section)

-        The Discussion section is often started with the main findings of the current study. Could you, please, consider this. For example, for comparison see the Bachert et al. Efficacy and safety of dupilumab in patients with severe chronic rhinosinusitis with nasal polyps, Lancet 2019.

Minor

-         There are two dots in the end of the last sentence of the Abstract. Please, remove the other one.

-         In the methods section ‘smoke’, should this be ‘smoking’ or ‘smoking habit’, please check the language.

Please, check the comments for the Authors. 

Author Response

According to reviewer 3 suggestions:

  1.      The numerical values of the main results have been reported in the abstract (page 1, lines 26-29)
  2. As suggested by the reviewer this is a retrospective study which involved two different centers. The nature of the study is reported in the text (page 2, lines 82 and 88).
  3.  Information on the use of nasal steroids was added to the method section as requested (pages 2 and 3, lines 99-102)
  4.  Information regarding OCS use is now reported in table 1 and results section (page 14, lines 284-286), as requested
  5.   Information on the use of asthma drugs is now added to the method section (page 3, lines 102-103), as requested
  6. The meaning of VAS-NO in the Methods section was explained as requested (page 3, line 124)
  7. The used variables have been added in the captions of the tables 2 and 3.
  8.  Information regarding the increased eosinophilia was added in the results section (page 14, lines 275-276, 281, 284-286)
  9.  The Discussion section is now starting with the main findings of the current study, as suggested
  10. We removed one of two dots in the end of the last sentence of the Abstract.
  11. In the methods section ‘smoke’ was substituted with ‘smoking habit’ (page 3, line 141)

Round 2

Reviewer 3 Report

It is written that ‘Multiple regression was fitted to consider all the variables examined (sex, age, smoking, asthma, endoscopic sinus surgery, the number of oral corticosteroids cycles over the past year, allergies, and NSAID intolerance or Widal’s disease triad) in a mixed effect model to test the significance of the differences between the older and adult patients over time.
Further it is written that ‘Multivariate linear regression analysis (linear mixed model) to assess the variable’s effect on VAS-NO. The last columns show the estimates of the effects and relative p-values for the univariate models.’

I found the presentation of multivariate and univariable logistic regression analysis confusing. In here dupilumab should be handled as the main exposure and VAS or SNOT as the main outcome and the other factors (age, gender, endoscopic surgery as independent or explanatory variables). Please, check how multivariable logistic logistic regression analysis was used in the Holguin F, et al. Obesity and asthma: an association modified by age of asthma onset. J Allergy Clin Immunol. 2011 Jun;127(6):1486-93.e2. doi: 10.1016/j.jaci.2011.03.036. PMID: 21624618; PMCID: PMC3128802.

or in the

Kole TM, et al. Predictors and associations of the persistent airflow limitation phenotype in asthma: a post-hoc analysis of the ATLANTIS study. Lancet Respir Med. 2023 Jan;11(1):55-64. doi: 10.1016/S2213-2600(22)00185-0. Epub 2022 Jul 27.

I suggest to either modify the Tables 2-5 or to leave them out of the manuscript.

Author Response

According to reviewer 3 suggestions, tables 2 and 3 have been modified following the example by Kole et al. Tables 4 and 5 have not been modified because they do not refer to the multivariate model.